# A Single Amino Acid Substitution in Porcine Reproductive and Respiratory Syndrome Virus Glycoprotein 2 Significantly Impairs Its Infectivity in Macrophages

**DOI:** 10.3390/v14122822

**Published:** 2022-12-18

**Authors:** Jayeshbhai Chaudhari, Raquel Arruda Leme, Kassandra Durazo-Martinez, Sarah Sillman, Aspen M. Workman, Hiep L. X. Vu

**Affiliations:** 1Nebraska Center for Virology, University of Nebraska-Lincoln, Lincoln, NE 68583, USA; 2School of Veterinary Medicine and Biomedical Sciences, University of Nebraska-Lincoln, Lincoln, NE 68583, USA; 3Clinal Research Department, Dechra Pharmaceuticals, Londrina 86030, Brazil; 4Department of Animal Science, University of Nebraska-Lincoln, Lincoln, NE 68583, USA; 5United State Department of Agriculture, Agriculture Research Service, U.S. Meat Animal Research Center, Clay Center, NE 68933, USA

**Keywords:** PRRSV, tropism, PAMs, GP2, CD163

## Abstract

Porcine reproductive and respiratory syndrome virus (PRRSV) has a restricted tropism for macrophages and CD163 is a key receptor for infection. In this study, the PRRSV strain NCV1 was passaged on MARC-145 cells for 95 passages, and two plaque-clones (C1 and C2) were randomly selected for further analysis. The C1 virus nearly lost the ability to infect porcine alveolar macrophages (PAMs), as well as porcine kidney cells expressing porcine CD163 (PK15-pCD163), while the C2 virus replicates well in these two cell types. Pretreatment of MARC-145 cells with an anti-CD163 antibody nearly blocked C1 virus infection, indicating that the virus still required CD163 to infect cells. The C1 virus carried four unique amino acid substitutions: three in the nonstructural proteins and a K160I in GP2. The introduction of an I160K substitution in GP2 of the C1 virus restored its infectivity in PAMs and PK15-pCD163 cells, while the introduction of a K160I substitution in GP2 of the low-passaged, virulent PRRSV strain NCV13 significantly impaired its infectivity. Importantly, pigs inoculated with the rNCV13-K160I mutant exhibited lower viremia levels and lung lesions than those infected with the parental rNCV13. These results demonstrated that the K160 residue in GP2 is one of the key determinants of PRRSV tropism.

## 1. Introduction

Porcine reproductive and respiratory syndrome virus (PRRSV) is the causative agent of a swine viral disease characterized by respiratory disorders in young pigs and reproductive failure in breeding pigs (reviewed in [1]). PRRSV belongs to the order *Nidovirales*, family *Arteriviridae*, and genus *Betaarteriviruses* [2]. PRRSV is classified into two major types: PRRSV-1 (*Betaarteriviruses suid 1*) and PRRSV-2 (*Betaarteriviruses suid 2*), which share approximately 60% nucleotide sequence identity [2]. PRRSV has a positive-sense, single-stranded RNA genome of about 15 kb in length, which is 5′ capped and 3′ end polyadenylated and contains at least fourteen open-reading frames (ORFs), eight of which encode for the viral structural proteins including: four enveloped glycoproteins (GP2, GP3, GP4, and GP5), three membrane proteins (Envelope (E), ORF5a, and Membrane (M) protein), and the Nucleocapsid (N) protein [3,4]. 

PRRSV has a restricted host and cell tropism. Pigs are the only known natural host for PRRSV. In vivo, the differentiated cells of monocyte and macrophage lineages such as porcine alveolar macrophages (PAMs), monocyte-derived dendritic cells (moDCs), tissue-resident macrophages, and pulmonary intravascular macrophages (PIMs) are the cellular targets for virus replication [5,6]. In vitro, PRRSV is mainly propagated on MA-104, the monkey kidney epithelial cells, and its derivative cell line, MARC-145 [7]. Several modified-live virus (MLV) vaccines for PRRSV have been generated by consecutively passaging the virus on MARC-145 cells [7]. Successive passaging of PRRSV on MARC-145 markedly reduces but does not completely abolish the virus infectivity on PAMs [8]. 

Cellular receptors are the main determinants of PRRSV tropism. At least seven receptors/attachment factors have been identified as PRRSV entry mediators including sialoadhesin (Sn or CD169) [9], CD151 [10], vimentin [11], DC-SIGN [12], heparan sulfate [13], MYH9 [14], and CD163 [15]. Of these receptors, Sn and CD163 are proposed to be the main receptors for PRRSV entry into macrophages. Sn interacts with the viral GP5-M heterodimer and facilitates virus attachment and internalization [16]. However, ectopic expression of Sn alone in a non-susceptible cell line such as the porcine kidney 15 (PK-15) does not render the cells susceptible to PRRSV [9]. Additionally, gene-edited pigs lacking Sn are fully susceptible to PRRSV infection, demonstrating that this receptor is not required for PRRSV infection [17]. Transfection of non-susceptible cell lines with CD163 renders the cells susceptible to PRRSV infection, while the treatment of PAMs and MARC-145 cells with an anti-CD163 antibody completely blocks the infection [15]. Additionally, gene-edited pigs lacking CD163 are resistant to PRRSV infection [18,19,20]. The viral GP2 and GP4 have been shown to interact with CD163 [21,22]. The substitution of ORFs 2–4 (encoding the GP2, GP3, GP4, and E protein) of a PRRSV strain by the corresponding ORFs of equine arteritis virus (EAV) altered the virus tropism where the chimeric virus lost the ability to infect PAMs while gaining the ability to infect baby hamster kidney-21 (BHK-21), the cell line that is normally used to propagate EAV [22].

Although it is well established that CD163 is a bona fide receptor for PRRSV entry, there is evidence that other unidentified cellular factors are required for productive PRRSV infection. Transfection of non-susceptible cell lines such as PK-15 and BHK-21 with CD163 cloned from human, monkey, dog, and mouse cell lines rendered the cells susceptible to PRRSV infection, but the original human and mouse cell lines from which the CD163 was obtained were not susceptible to PRRSV [15]. This observation leads to the hypothesis that unknown cellular factors that are required for productive PRRSV infection might be present in cell lines such as BHK-21 and PK-15 but absent in mouse and human macrophages. Although the expression of porcine CD163 alone in PK-15 cells is enough to render the cells susceptible to PRRSV, the level of infection can be greatly enhanced when the cells are co-transfected with CD163 and Sn [23]. Additionally, receptor usage varies greatly among PRRSV strains with different levels of virulence. Low virulent PRRSV isolates mainly infected cells expressing both Sn and CD163 (Sn^+^CD163^+^) while highly virulent PRRSV strains can infect multiple cell types including Sn^+^CD163^+^, Sn^−^CD163^+^, and to a lesser extent Sn^−^CD163^−^ [24].

We were interested in developing a new live-attenuated PRRSV vaccine candidate. For this purpose, the PRRSV-2 isolate designated as NCV1 was passaged 95 times on MARC-145 cells, followed by three consecutive rounds of plaque purification. Two plaque-purified clones (C1 and C2, respectively) were randomly selected to evaluate their infectivity on PAMs. Surprisingly, the C1 virus exhibited a significant impairment in its infectivity in PAMs and PK15 cells expressing porcine CD163 (PK15-pCD163) while the C2 virus maintained the infectivity in these two cell types. By using reverse genetics, we demonstrated that the amino acid residue K160 in GP2 is a key determinant of PRRSV infectivity in PAMs. 

## 2. Materials and Methods

### 2.1. Cells, Antibodies, and Reagents

MARC-145, a monkey kidney cell line, was cultured in Dulbecco’s Modified Eagle Medium (DMEM) containing low glucose and low bicarbonate. PK15-pCD163 (kindly provided by Dr. X.J Meng at Virginia Polytechnic Institute and State University) and HEK-293T cells were maintained in DMEM high glucose. PAMs used for the ex vivo infections were isolated from the lung lavage of pigs between 4- or 8-weeks-old. All media were supplemented with 10% fetal bovine serum (FBS, Sigma, St. Louis, MO, USA), 100 units/mL of penicillin, and 100 μg/mL of streptomycin (Sigma, St. Louis, MO, USA). All cell types used in this study were cultured at 37 °C, with 5% CO_2_. The mouse monoclonal antibody SDOW17 specific to PRRSV N protein was purchased from the National Veterinary Services Laboratories (Ames, IA, USA). Alexa Fluor-488 F(ab’)2 fragment of goat anti-mouse IgG (H+L) antibody was purchased from Invitrogen (Eugene, OR, USA). Fluorescein isothiocyanate (FITC) conjugated mouse anti-PRRSV N protein antibody (clone SR30) was purchased from Rural Tech lnc., (Brookings, SD, USA). Goat anti-human CD163 polyclonal antibody was purchased from R&D system (clone AF1407, Minneapolis, MN, USA). Goat anti-mouse IgG (H+L)-HRP antibody and DAPI (4′,6-diamidino-2-phenylindole dihydrochloride) were purchased from ThermoFisher Scientific (Carlsbad, CA, USA).

### 2.2. Viruses and Full-Length Infectious cDNA Plasmids

The PRRSV strain NCV1 (GenBank no. ON950548) was isolated from a serum sample collected from a 3-week-old piglet from a farm that was affected with the PRRSV [25]. The virus was passaged on MARC-145 cells for 95 passages, followed by three consecutive rounds of plaque purification. Two random plaques were picked and amplified for two additional passages in MARC-145 cells to have enough virus stocks for further studies. The construction of the NCV1 C1 infectious cDNA clone (designated pNCV1 in our previous study [25] or designated pC1 in this study) has been previously described. 

The PRRSV strain NCV13 (GenBank no. KX192112) was isolated in 2014 from a serum sample of a pig that was infected with PRRSV [26]. After two passages in PAMs, the virus was passaged in MARC-145 cells for a total of 12 passages. To construct the infectious cDNA clone pNCV13, viral RNA was isolated from the NCV13 passage 12th using the QIAmp viral RNA isolation kit (Qiagen GmbH, Hilden, Germany). Four overlapping PCR amplicons encompassing the full-length sequences of the NCV13 genome were amplified using the primers listed in Table 1. The amplicons were sequentially assembled into a bacterial plasmid immediately downstream of the human cytomegalovirus (hCMV) early promoter. The hepatitis delta virus ribozyme sequence was incorporated immediately downstream of the viral poly A tail to ensure the authenticity of the viral genome 3′ terminus. The resulting full-length cDNA clone was subjected to DNA sequencing to ensure sequence authenticity.

### 2.3. Site Directed Mutagenesis in GP2 and Recovery of PRRSV Strains

Site-directed mutagenesis was carried out using synthetic primers (Table 2) to introduce the I160K substitution in the GP2 of the pC1 cDNA clone and K160I substitution in the GP2 of the pNCV13 cDNA clone. To construct the pC1-I160K plasmid, two overlapping PCR fragments containing 25nt were generated. The first PCR fragment, spanning from the enzyme site EcoRV to the aa 160 in GP2 was generated using primer pair 11340F and pC1-I160K-R. The second PCR fragment, spanning from aa 160 in GP2 to the enzyme site XbaI in GP5 was generated using primer pair pC1-I160K-F and P14461R. These two overlapping fragments were then fused together to generate a fragment spanning from the EcoRV to XbaI enzyme sites and cloned in to pC1 backbone. A similar strategy was used to construct the pNCV13-K160I cDNA clone. The resulting full-length cDNA clones were subjected to DNA sequencing to ensure sequence authenticity.

The recovery of the infectious virus from full-length cDNA clones was carried out as described previously [25]. Briefly, MARC-145 cells were seeded in a 6-well plate at the cell density of 3 × 10^5^ per well for 24 h. The cells were then transfected with 2.5 μg of plasmid using the Trans-IT-X2 transfectant (Mirus Bio LLC, Madison, WI, USA) as per the manufacturer’s recommendations. At 96 h post-transfection (hpt) or when an obvious cytopathic effect was observed, cell culture supernatant was harvested and further propagated in MARC-145 for up to four passages to obtain enough virus stock for further studies. The viruses were stored in small aliquots at −80 °C. 

### 2.4. Viral Genome Sequencing

The complete genome of viruses used in this study were sequenced as previously described [26]. Total RNA was purified either from the viral culture supernatant collected from the MARC-145 infected cells or serum of infected pigs using the Quick-RNA^TM^ Viral Kit (Zymo research, Costa Mesa, CA, USA) following the manufacturer’s recommendation. Sequencing libraries were prepared using the Illumina TruSeq RNA Kit and sequenced with 2 X 300 paired-end reads on the Miseq platform (Illumina, San Diego, CA, USA). Assembly of the viral genomes was performed as previously described [26]. 

### 2.5. Generation PK15 Cells Expressing Porcine or Monkey CD163

Total RNA was extracted from the PAMs or MARC-145 cells using the Direct-zol^TM^ RNA MicroPrep (Zymo research, Costa Mesa, CA, USA) following the manufacturer’s recommendation. cDNA synthesis was performed using oligo-dT primer and the SuperScript^TM^ IV reverse transcriptase kit (Invitrogen, Vilnius, Lithuania) as per the manufacturer’s recommendations. Porcine or monkey CD163 gene fragments were then PCR amplified using the primer pair listed in Table 3 and subsequently cloned into the pHASE-HYG-MCS vector using the NotI-NheI enzyme site.

The three-plasmid lentivirus system was used to produce recombinant lentivirus. HEK293T cells were transfected with pHASE-HYG-porcineCD163 or pHASE-HYG-monkeyCD163 vector plasmid in combination with two packaging plasmids, pVSVG and pGAG/Pol. At 16 hpt, the medium was replaced with the fresh medium containing 5 mM sodium butyrate. At 24 hpt, the medium was again replaced with the fresh medium containing 10 mM HEPES. At 48 hpt, the supernatant containing recombinant lentiviruses was harvested and polybrene was added at 10 μg/mL and filtered using the 0.45 μm filter. Recombinant lentiviruses were then aliquoted and stored at −80 °C. 

For the generation of the PK15 cell line expressing CD163 receptor, cells were seeded at a density of 3 × 10^5^ cells per well of the 6-well plate. At 24 h post seeding, cells were transduced with the lentiviruses carrying either the porcineCD163 or monkeyCD163 transgene. Transduced cells were selected with 100μg/mL hygromycin B (Mirus Bio LLC, Madison, WI, USA).

### 2.6. Multi-Step Growth Curve Analysis 

To evaluate the virus growth kinetics, MARC-145 cells were cultured in a 24-well plate at a density of 3 × 10^4^ cells per well. After 48 h post-seeding, cells were infected with different PRRSV strains at the multiplicity of infection (MOI) of 0.1 TCID50 per cell. After 1 h of adsorption, cells were washed thrice using DMEM to remove the unbound virus and supplemented with 500μL fresh cDMEM. Culture supernatant was harvested at 0, 12, 24, 36, 48, 60, and 72 h post-infection (hpi) and stored at −80 °C. Infectious virus titers were determined by plaque assay on MARC-145. 

### 2.7. Indirect Immunofluorescence Assay 

Indirect immunofluorescence assay (IFA) was performed using the SDOW-17 monoclonal antibody specific to the viral N protein as previously described [25]. Briefly, infected cells in the 24-well plates were washed twice with PBS (pH 7.4), fixed with a cold solution of methanol: acetone (1:1 *v*/*v*) for 10 min, followed by an air-dry. The cell monolayer was washed one time with PBS and incubated with the SDOW17 antibody (1:500 dilution in PBS) for 1 h at RT, followed by three washes with PBS. Subsequently, the cells were incubated with an Alexa Fluor-488 F(ab’)_2_ fragment of goat anti-mouse IgG antibody (diluted 1:1000 in PBS) for 1 h at RT, followed by three washes with PBS. Nuclear staining was performed using DAPI (4′, 6′-diamidino-2-phenylindole) diluted 1:4000 in PBS for 7 min at RT. After three washes with PBS, a fluorescent signal was observed under an inverted fluorescence microscope. Fluorescent images were taken using a Nikon Eclips Ts2R-FL operated by Nikon NIS Elements (ver 5.02). All images were taken separately with either 10× or 20× objectives using green or blue filters. Images for each filter channel were then overlaid to generate the final overlapped images.

### 2.8. Flow Cytometry

The frequencies of cells infected with PRRSV were quantified using flowcytometry as previously described [27]. Cryopreserved PAMs were revived and cultured in cRPMI in flowcytometry tubes and were inoculated with PRRSV strains at MOI of 0.1 or 10 TCID50 per cells. After 1 h adsorption at 37 °C, the inoculum was removed, and the cells were washed thrice with the culture medium and supplemented with 500μL of cRPMI-1640 medium. At 12 (10 MOI) or 24 (0.1 MOI) hpi, the cells were collected and immunostained with an anti-N antibody SR30-FITC. In the case of PK15-pCD163, cells cultured in 24-well plates were inoculated with PRRSV strains at different MOIs. At 72 hpi, the cell monolayer was treated with trypsin to make single cell suspension before staining. Cells were first stained using Zombie Violet Live-dead fixable dye (BioLegend, San Diego, CA, USA) (dilution 1:100 in PBS) for 20 min at RT in the dark. Fixation was performed using 4% paraformaldehyde (Bio-Rad, Hercules, CA, USA) as per the manufacturer’s recommendations. The cells were then incubated with the SR30-FITC monoclonal antibody diluted 1:100 in Leucoperm buffer (Bio-Rad, Hercules, CA, USA) for 30 min on ice in the dark. After three washes with the FACS buffer, the cells were analyzed by using a CytoFlex cytometer (Beckman Coulter, Fremont, CA). Mock-infected cells were used to decide gating. Approximately 30,000 events were acquired for each sample and collected data were analyzed using the FlowJo software (BD Biosciences, San Jose, CA, USA).

### 2.9. CD163 Receptor Blocking Assay

MARC-145 and PAM cells were seeded in a 48-well plate at the cell density of 5 × 10^4^ per well. One day later, the cells were first incubated either with 10 μg of goat anti-human CD163 polyclonal antibody (R&D System, Minneapolis, MN, USA) or normal goat IgG diluted in 100μL of culture medium for 1 h at 37 °C. Subsequently, the treated cells were inoculated with PRRSV strains at an MOI of 2.5 TCID50 per cell. After 1 h adsorption at 37 °C, the inoculum was removed, and cells were washed thrice with the culture medium and supplemented with 500 μL of cDMEM or cRPMI-1640 medium. At 12 hpi, the culture supernatants were collected to determine the viral genome copy by RT-PCR and the cells were analyzed by IFA and flowcytometry to quantify the percentage of PRRSV-infected cells. 

### 2.10. Animal Studies

Eighteen 3-week-old, PRRSV-negative pigs were randomly divided into three groups with an equal number of male and female piglets per group. After 1 week of acclimation, group 1 was injected with 2 mL DMEM to serve as the non-infection control. Group 2 was inoculated with the rNCV13 virus while group 3 was injected with the rNCV13-K160I mutant, intramuscularly with 2 mL inoculum containing 10^5.0^ TCID_50_. Whole-blood samples were collected at 0, 4-, 8-, 11-, and 14- dpi and serum was isolated to measure viral RNA copy and antibody response. At 14 dpi, all pigs were humanely euthanized. During necropsy, samples of lung were collected and fixed in 10% neutral buffered formalin and processed by routine procedures for histopathologic examination. Histological evaluation of the tissue sections was performed by a board certified veterinary pathologist who was blinded to the experimental design. Lung sections were evaluated for interstitial pneumonia and lung consolidation. Severity was scored as follows: 0, normal; 1, mild; 2, moderate multifocal; 3, moderate diffuse; 4, severe. RNA in situ hybridization (ISH) was performed to detect PRRSV mRNA transcript in lung tissue sections as previously described (23).

### 2.11. Measurement of Viral RNA Copy Number in Culture Supernatant and Serum 

Total RNA was extracted from the culture supernatant or serum samples using a viral RNA minikit (Quiagen GmbH, Hilden, Germany) following the manufacture’s recommendation. Viral copies were quantified using a commercial reverse transcriptase quantitative PCR (RT-qPCR) kit (Tetracore. Inc., Rockville, MD, USA) and was reported as log_10_ viral genome copies per ml of total RNA used in the RT-qPCR. For statistical purposes, samples that have no detectable levels of viral RNA were assigned a value of 0 log_10_ copies. In the case of the culture supernatant, to calculate the difference of viral genome copy numbers, viral genome copies detected at 0 hpi were subtracted from 24 hpi.

### 2.12. Statistical Analysis

Statistical analyses were performed using GraphPad Prism v 8.4.3 (GraphPad Software Inc.). Comparisons of mean ± SE of two or more groups at one time point were analyzed using a one-way analysis of variance (ANOVA) followed by Dunnett’s multiple comparisons test, respectively. The mean ± SE of two or more groups at multiple time points was analyzed using a two-way ANOVA followed by Sidak’s multiple comparison test. For all comparisons, a *p*-value < 0.05 was considered significant.

## 3. Results

### 3.1. Identification of a Plaque-Clone of a High-Passage PRRSV Strain with Impaired Infectivity in PAMs 

The adaptation of PRRSV on MARC-145 cells reduces the virus’ ability to infect PAMs [28]. Therefore, we were interested in comparing the infectivity of the parental P2 virus and the high-passage NCV1 (P95) and the plaque-purified clones (C1 and C2) in PAMs. An infectivity assay was conducted using PAMs collected from four piglets between 4- and 8-weeks-old to account for the influence of host genetics on the susceptibility to PRRSV infection [29]. PAMs were infected with P2, P95, C1, and C2 viruses at an multiplicity of infection (MOI) of 0.1 tissue culture infectious dose 50 (TCID_50_) per cell. As expected, the numbers of PAMs expressing the viral N-protein (N^+^) were significantly lower in cells inoculated with the P95 and C2 virus than cells inoculated with the P2 virus (Figure 1A). Interestingly, only a few N+ cells were detected in PAMs inoculated with the C1 virus. Quantitatively, the frequencies of N^+^ cells detected in PAMs inoculated with the P95 and C2 viruses were 2.7% and 1.2%, respectively, which were significantly lower than the frequency of N^+^ detected in PAMs inoculated with the parental P2 virus (81.2% Figure 1B,C). Only 0.16% of N^+^ cells were detected from PAMs inoculated with the C1 virus. Accordingly, the viral genome copies and infectious virus titers were significantly lower in PAMs inoculated with the C1 virus (Figure 1D,E). The results suggested that the C1 virus had a severe defect in the ability to infect PAMs. 

To further evaluate the C1 virus infectivity in PAMs, the cells were inoculated at an MOI of 10 TCID_50_ per cell. Because the parental P2 virus is not well adapted to MARC-145 cells, high virus titer was not achieved for the P2 virus; therefore, we excluded parental P2 from the high MOI studies. At this high infection dose, the frequencies of N^+^ cells observed in PAM inoculated with the P95 and C2 viruses increased to 61.9% and 54.5%, respectively, while only 6.5% of N^+^ cells were detected in PAMs inoculated with the C1 virus (Figure 2A,B). The viral genome copies were numerically lower in supernatants from PAMs inoculated with the C1 virus (Figure 2C) while the infectious virus titer was significantly lower in supernatants collected from PAMs inoculated with the C1 virus (Figure 2D). Collectively, the results clearly demonstrated that the C1 virus significantly lost its ability to infect PAMs.

### 3.2. The C1 Virus Exhibited Significantly Lower Infectivity in PK-15 Cell Line Stably Expressing Porcine CD163 

CD163 is a key receptor for PRRSV entry into PAMs and the ectopic expression of CD163 is sufficient to render the non-permissive cell line such as PK-15 susceptible to PRRSV infection [30]. Since the C1 virus exhibited a significant impairment in the ability to infect PAMs, we wanted to determine the virus infectivity in the PK-15 cell line stably expressing porcine CD163 (PK15-pCD163). At 72 hpi, only a few small foci of N^+^ cells were observed in PK15-pCD163 cells inoculated with the C1 virus while multiple large foci of N^+^ cells were observed in cells inoculated with the P95 and C2 virus (Figure 3A). On average, 1.13% of N^+^ cells were detected in PK15-pCD163 cells inoculated with the C1 virus, which was significantly lower than the frequencies of N^+^ cells detected in cells inoculated with P95 (7.2%) or C2 (4.7%) (Figure 3B). Accordingly, the viral genome copies and infectious virus titers were also significantly lower in the PK15-pCD163 cells inoculated with the C1 virus than those inoculated with the P95 and C2 viruses (Figure 3C,D). It is noteworthy that the P95, C1, and C2 viruses replicated efficiently in MARC-145 cells, with similar growth kinetics (Appendix A). 

### 3.3. Comparative Analysis of the C1 Virus Genome Sequence 

The full genomes of the NCV1 P2, P95, C1, and C2 viruses were sequenced and aligned to identify potential substitutions that might be responsible for the impaired infectivity of the C1 virus in PAMs. Compared to the parental P2 virus, the P95, C1, and C2 viruses carried, respectively, 55, 66, and 58 nucleotide changes, which were randomly distributed throughout the viral genomes (Appendix A and Figure 4A). Regarding amino acid (aa) changes, the P95, C1, and C2 viruses carried 24, 25, and 26 aa substitutions, respectively. Of the 25 amino acid substitutions detected in the C1 virus, 21 were shared with either the P95 or C2 viruses (Table 4). Four aa substitutions that were only found in the C1 virus but not P95 or C2 included three substitutions in nonstructural proteins (A1336S, D2080N, and N3496D) and one in GP2 (K160I) (Figure 4A and Table 4).

### 3.4. An I160K Substitution in the C1 GP2 Restored the Virus Infectivity in PAMs and PK15-pCD163 Cells

Since GP2 has been demonstrated to interact with CD163 [21], we focused on determining the potential effects of the GP2 K160I substitution on the C1 virus infectivity. Analysis of 1467 PRRSV-2 GP2 sequences available in GenBank revealed that they all carry a K residue at position 160 in GP2, indicating that this aa position is highly conserved among PRRSV-2 isolates (Figure 4B). 

Using an infectious cDNA clone of the C1 virus (designated as pC1), site-directed mutagenesis was used to introduce an I160K substitution in GP2 to generate the pC1-I160K cDNA clone. Recombinant viruses rescued from the pC1 and pC1-I160K cDNA clones were designated rC1 and rC1-I160K viruses, respectively. The rC1 and rC1-I160K viruses exhibited similar growth kinetics on MARC-145 cells (Appendix A). When tested in PAMs, the rC1 virus exhibited a similar infection phenotype as of its parental strain, the C1 virus (Figure 4C). The frequency of N^+^ cells was significantly higher in PAMs inoculated with the rC1-I160K mutant than with rC1 or C1 viruses (19.22% vs. 1%) (Figure 4D). Notably, the frequency of N^+^ cells in PAMs inoculated with the rC1-I160K mutant was not significantly different than in PAMs inoculated with the C2 virus. The viral genome copies in culture supernatant from PAMs inoculated with the rC1-I160K mutant was not statistically different than the C2 virus but was significantly higher than the rC1 or C1 viruses (Figure 4E). 

When tested in PK15-pCD163 cells, the rC1-I60K mutant exhibited a similar infectivity pattern as compared to the C2 virus, with multiple large foci of N^+^ cells detected at 72 hpi (Figure 5A). On the other hand, both the rC1 and C1 viruses had only a few single-cell foci detected. The frequency of N^+^ cells as well as the viral genome copies was significantly higher in PK15-pCD163 cells inoculated with rC1-I160K than with rC1 (Figure 5B,C). Altogether, the results demonstrated that the introduction of the I160K substitution in GP2 of the C1 virus significantly restored the virus’ ability to infect PAMs and PK15-pCD163 cells. 

### 3.5. CD163 Is Necessary but Not Sufficient for the C1 Virus Infection

Since the C1 virus did not efficiently infect PAMs or PK15-pCD163, we sought to determine if it still required CD163 for infection. A receptor blocking experiment was conducted using a polyclonal antibody (pAb) specific to human CD163. This anti-CD163 antibody has been shown to effectively block PRRSV infection of MARC-145 cells in several previous studies [15]. Pretreatment of MARC-145 cells with the anti-CD163 pAb significantly reduced the number of cells being infected with C2, rC1, and rC1-K160I viruses (Figure 6A). Interestingly, pretreatment of MARC-145 cells with the normal goat IgG slightly enhanced the frequencies of cells being infected with all three viruses tested (Figure 6B,C). Collectively, the results demonstrated that the C1 virus was still used on CD163 to infect MARC-145 cells.

To this end, our data demonstrated that the rC1 virus infectivity in MARC-145 cells was abolished when the cells were pretreated with the anti-CD163 pAb (Figure 6), but the virus did not efficiently infect PK15-pCD163 or PAMs (Figure 4 and Figure 5). Therefore, we hypothesized that the C1 virus, after being passaged in MARC-145 cells for 95 passages, adapted to preferentially use monkey CD163 instead of porcine CD163. To test this hypothesis, we generated a lentivirus expressing either porcine CD163 (pCD163) or monkey CD163 (mCD163) and transduced PK15 cells in the same manner to ensure that the expression levels of pCD163 and mCD163 were similar (Appendix A). Only a few N^+^ cells were detected in mCD163-expressing cells inoculated with the rC1 virus whereas multiple large foci of N^+^ cells were detected in mCD163-expressing cells inoculated with the rC1-I160K mutant (Figure 7A). The frequencies of N^+^ cells were not statistically different between mCD163- and pCD163-expressing cells, regardless of whether they were inoculated with the rC1 or rC1-I160K mutant (Figure 7B). The viral genome copies were significantly higher in pCD163- and mCD163-expressing cells inoculated with rC1-I160K than those inoculated with rC1 (Figure 7C). Collectively, the results demonstrated that the rC1 virus did not preferentially use monkey CD163 instead of the porcine CD163 receptor. 

### 3.6. A K160I Substitution in GP2 of a Low-Passage PRRSV Strain Impaired Its Infectivity in PAMs and PK15-pCD163 Cells

Since the GP2 K160 residue is highly conserved among PRRSV-2 isolates (Figure 4B), we wanted to investigate how this aa residue affects the infectivity of other PRRSV-2 strains. To this end, an infectious cDNA clone of the low-passage PRRSV strain NCV13 was constructed, and side-directed mutagenesis was used to introduce a K160I substitution in its GP2. Sequencing of the whole genome of the resulting mutant virus (rNCV13-K160I) identified two unwanted substitutions in nsp2: V462A and T718A. These two substitutions were found in sequences of PRRSV-2 field isolates deposited in GenBank (Appendix A). Particularly, the A462 and A718 residues in nsp2 were, respectively, found in 14.67% and 10.67% of the PRRSV field isolates analyzed (Appendix A). We therefore assumed that these two unwanted substitutions should not affect the virus infectivity in PAMs or PK15-pCD163 cells.

The rNCV13-K160I mutant exhibited a slight reduction in its replication in MARC-145 cells as compared to the parental strain rNCV13 (Appendix A). When tested in PAMs, the rNCV13-K160I mutant exhibited a marked reduction in infectivity as compared to the parental strain rNCV13 (Figure 8A). On average, only ~1.9% of N^+^ cells were detected in PAMs inoculated with the rNCV13-K160I mutant while ~42.6% of N^+^ cells were detected in PAMs inoculated with the rNCV13 virus (Figure 8B,C). Likewise, the viral genome copy was also significantly lower in PAMs inoculated with the rNCV13-K160I mutant than with the rNCV13 (Figure 8D).

The rNCV13-K163I mutant also exhibited a marked reduction in its infectivity in PK15-pCD163 cells. While only a few small foci of N^+^ cells were detected in PK15-pCD163 inoculated with the rNCV13-K160I mutant, multiple large foci of N^+^ cells were readily detected in cells inoculated with the rNCV13 (Figure 9A). Quantitatively, the frequency of N^+^ cells was significantly lower in PK15-pCD163 inoculated with the rNCV13-K160I mutant than those inoculated with the rNCV13 (1.6% vs. 0.55%) (Figure 9B). Accordingly, the viral genome copy was significantly lower in PK15-pCD163 inoculated with the rNCV13-K160I mutant than with the rNCV13 (Figure 9C). Together, the results demonstrated that the K160I substitution in GP2 of the low-passage PRRSV-2 strain rNCV13 significantly impaired the virus infectivity in PAMs and PK15-pCD163 cells. 

### 3.7. A K160I Substitution in GP2 of a Low-Passage PRRSV Strain Reduced the Virus Replication in Pigs

Since the rNCV13-K160I mutant exhibited a significant impairment in its infectivity in PAMs cultured ex vivo, we sought to determine the infectivity of this mutant virus in pigs. Six pigs were inoculated with the rNCV13-K160I mutant and all of them became viremic. However, at 4- and 8-days post-infection (dpi), the viremia levels in pigs infected with the rNCV13-K160I mutant were approximately 1 log_10_ lower than those infected with the parental rNCV13 strain (Figure 10A). No significant difference in viremia levels was observed at later time points. All pigs infected with the rNCV13-K160I mutant seroconverted at 14 dpi, and the antibody levels were not statistically different from pigs infected with rNCV13 (Figure 10B). 

Regarding the lung pathology, pigs infected with the rNCV13-K160I mutant exhibited milder microscopic lesions than those infected with rNCV13 (Figure 10C,E). Additionally, there was a slightly lower frequency of virus-infected cells in lung sections from pigs infected with the rNCV13-K160I mutant than those infected with rNCV13 (Figure 10C,D). Collectively, the results demonstrated that the rNCV13-K160I mutant was able to infect pigs and induced lung lesions, but the level of virulence was significantly lower than the parental rNCV13 strain. 

To determine the genetic stability of the rNCV13-K160I mutant virus when replicating in pigs, RNA extracted from serum samples collected at 4- and 14- dpi from one pig that exhibited the highest viremia levels were used for the whole viral genome sequencing. RNA sequences of rNCV13-K160I mutant at 4- and 14- dpi did not have any substitutions compared to the sequence of the input virus, indicating that the rNCV13-K160I mutant was stable during virus replication in pigs (Figure 10F). Therefore, the enhanced levels of viremia in pigs inoculated with rNCV13-K160I observed at later time points after inoculation was not due to the reversion to wild-type sequence.

## 4. Discussion

The propagation of viruses in cell cultures often results in substitutions in the viral genomes, which sometimes alter the viral tropism. For instance, field isolates of foot-and-mouth disease virus (FMDV) require RGD-dependent integrin receptors to infect epithelial cells. In contrast, cell culture-adapted FMDV variants acquire substitutions in the capsid proteins, which allow them to use heparin-sulfate as the receptor, and no longer require integrin for infection [31]. Human rhinovirus (HRV89) adapted in Hep-2 cells can infect the COS-7 cells that lack its entry receptor ICAM-1 [32]. Laboratory-adapted Ebola virus (EBOV) strains acquire substitutions in their glycoprotein, which enhance their replication in monkey and human cell lines [33]. PRRSV is known to infect pigs only, and macrophages residing in lung and lymphoid organs are the main targets for virus replication. MARC-145 cells are widely used to propagate PRRSV in vitro. However, PRRSV field isolates often need to be adapted through multiple passages before they can grow effectively in this non-host cell line [34]. Successive passage of virulent PRRSV strains in MARC-145 cells for about 80 passages or more significantly reduces, but does not totally impair, the virus infectivity in PAMs [28]. In the present study, we successively grew the PRRSV strain NCV1 for 95 passages in MARC-145 cells, followed by three consecutive rounds of plaque purification. Consistent with previous studies, we observed that NCV1 P95 exhibited a significant reduction in its ability to infect PAMs as compared to the low-passage NCV1 P2 (Figure 1). Interestingly, we discovered one plaque-clone (C1) that nearly lost the infectivity on PAMs, especially at a low MOI (Figure 1). Compared to the P95 and C2 viruses, the C1 virus has four unique aa substitutions: three in non-structural proteins and one in GP2. Introduction of the I160K substitution in GP2 of the rC1 virus restored its infectivity in PAMs and PK15-pCD163 cells to levels closely similar to that of the C2 clone. On the other hand, introduction of a K160I substitution in GP2 of the low-passage PRRSV strain NCV13 significantly reduced the virus infectivity in PAMs and PK15-pCD163 cells. Collectively, our results demonstrate that the residue K160 in GP2 contributes significantly to PRRSV-2 infectivity in PAMs.

CD163 is an important cellular receptor for PRRSV entry into macrophages [15]. Blockage of the CD163 receptor by an anti-CD163 antibody completely prevents susceptible cells from being infected with PRRSV [15,23]. Additionally, gene-edited pigs lacking CD163 are resistant to PRRSV infection [18]. However, cells expressing CD163 are not necessarily susceptible to PRRSV. While PAMs collected from young and adult pigs express similar levels of CD163, PAMs collected from young pigs are more susceptive to PRRSV infection than those collected from adult pigs [29]. Transfection of a mouse or human CD163 into PK15 cells renders the cells susceptible to PRRSV infection, but the original mouse and human cells from which the CD163 gene was obtained to transfect PK15 were not susceptible to PRRSV [15]. Thus, other cellular factors besides CD163 are required for productive PRRSV infection. Consistent with previous studies, we observed that pretreatment of MARC-145 cells with an anti-CD163 antibody completely prevents the cells from being infected with the C1 virus (Figure 6), clearly indicating that the virus requires CD163 for infection. On the other hand, the C1 virus does not infect PK15 cells expressing either porcine or monkey CD163 (Figure 7), thus ruling out the possibility that it has adapted to preferentially use monkey CD163 over porcine CD163. Together, the results indicate that CD163 is necessary but not sufficient for the C1 virus infection. Thus, the C1 virus productive infection might depend on cellular factors that are present in MARC-145 cells but not in PAMs or PK15 cells. Further studies are needed to identify those factors. 

Infectivity in PAMs cultured ex vivo has been proposed as a parameter to predict the virulence of PRRSV strains [28]. The introduction of a K160I substitution in GP2 of the NCV13 significantly impaired its replication in PAMs cultured ex vivo. Pigs inoculated with the rNCV13-K160I mutant exhibited lower levels of viremia than those inoculated with the parental rNCV13 virus at 4- and 7-dpi but not at the later time points. Additionally, pigs inoculated with the rNCV13-K160I mutant exhibited a significantly lower lung lesion score than those inoculated with the rNCV13 virus, but higher than non-infected pigs (Figure 10). RNA sequences of the rNCV13-K160I virus from blood samples collected at 4- and 14- dpi exhibit no substitutions as compared to the input virus RNA sequence. Together, the results demonstrate that the rNCV13-K16I mutant is only partially attenuated when tested in a young pig model, although it exhibits a marked reduction in its infectivity in PAMs cultured ex vivo. We postulate that the rNCV13-K160I mutant might be able to infect different cell types besides PAMs when inoculated into pigs. We observed in our previous studies that pigs inoculated intramuscularly with a wild-type PRRSV strain exhibited high levels of viremia within 24 hpi when the virus had not yet reached the lung to infect PAMs [27]. Additionally, viremia levels were relatively high at 15 dpi, when the frequency of PRRSV^+^ PAMs declined to less than 0.04%. Indeed, PRRSV^+^ cells can be detected in multiple tissues of pigs inoculated with virulent PRRSV strains, including PIMs, resident macrophages, and potentially dendritic cells in lymphoid tissues [5,6]. Studies on pig nasal mucosa explants revealed that PRRSV antigens can be detected in cells that do not express CD163, suggesting that the virus might infect cells independent of this receptor [24]. Although PRRSV can infect different cell types in pigs, most ex vivo studies related to the mechanisms of PRRSV cellular entry have been done on PAMs since they are relatively easy to harvest. The results of this study suggest that it is important to study the entry mechanisms and cellular factors required for PRRSV infection in different cell types besides PAMs to further understand viral pathogenesis.

## 5. Conclusions

In summary, we discovered a highly conserved aa residue in GP2 of PRRSV-2 that plays an important role in viral tropism. Particularly, PRRSV strains with a K160 in GP2 efficiently infect MARC-145, PK15-pCD163, and PAMs while mutants with a K160I substitution in GP2 infect MARC-145 but not PK15-pCD163 and PAMs, even though this mutant still requires CD163 for infection. Importantly, a low-passage PRRSV mutant carrying an I160 in GP2 exhibited reduced virulence compared to its parental strain. Additional studies are needed to understand how virus infectivity in PAMs is affected by the K160I substitution and identify potential cellular factors that control PRRSV infectivity in PAMs and other cell types. A better understanding of the molecular determinants of PRRSV tropism will be useful for the molecular attenuation of PRRSV, as well as to generate PRRSV-resistant pigs, either through breeding selection or gene editing. 

## Figures and Tables

**Figure 1 viruses-14-02822-f001:**
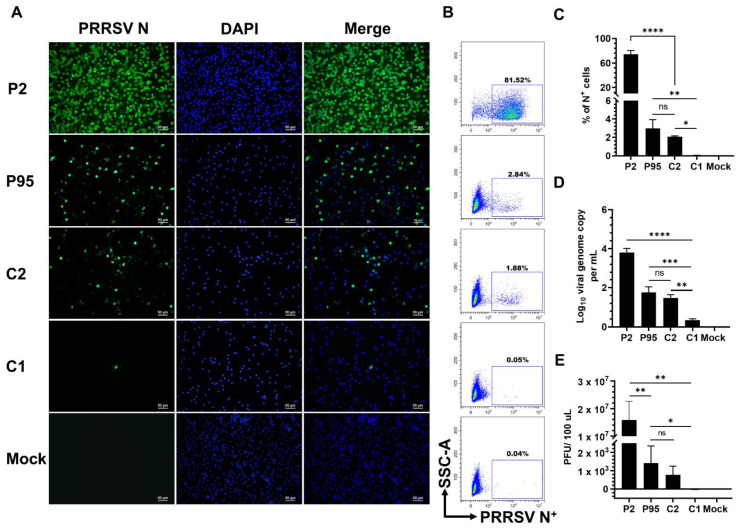
Infectivity of NCV1 P95 and its derived clones in PAMs at low MOI. PAMs collected from four piglets were inoculated with the indicated viruses at an multiplicity of infection (MOI) of 0.1 tissue culture infectious dose 50 (TCID_50_) per cell. All data were collected at 24 hpi. (**A**) Representative images of cells that were fixed and stained with an anti-N antibody to visualize virus-infected cells (green color). DAPI was used to stain the nuclei (blue color). Scale bar = 50 μm. (**B**) Representative flowcytometry plots of PAMs that were fixed and stained with a FITC-labeled anti-N antibody. Uninfected cells were used to set-up gating. (**C**) Percentage of PAMs expressing N protein (N^+^) as quantified by flowcytometry. (**D**) Difference of viral genome copy numbers in the culture supernatants collected at 0 hpi and 24 hpi as quantified by RT-qPCR. (**E**) Difference of virus infectious titers in the culture supernatants collected at 0 hpi and 24 hpi as quantified by plaque assay in MARC-145 cells. Data are presented as means and SEM of the three independent experiments each of which involved four populations of PAMs collected from four different pigs. ns: non-statistically significant, * *p* < 0.05, ** *p* < 0.01, *** *p* < 0.001, **** *p* < 0.0001.

**Figure 2 viruses-14-02822-f002:**
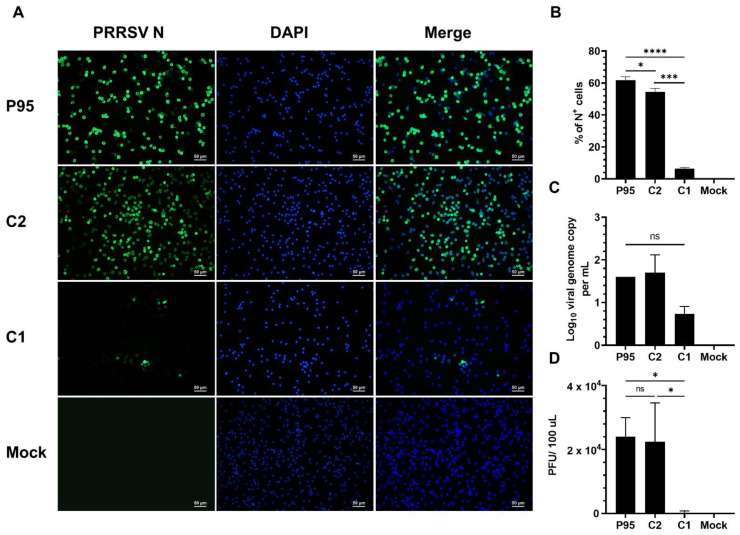
Infectivity of NCV1 P95 and its derived clones in PAMs at high MOI. PAMs collected from four piglets were inoculated with the NCV1 P95 and its derivatives C1 and C2 at an MOI of 10. All data were collected at 12 hpi. (**A**) Representative images of cells that were fixed and stained with an anti-N antibody to visualize virus-infected cells (green color). DAPI was used to stain the nuclei (blue color). Scale bar = 50 μm. (**B**) Percentage of PAMs expressing N protein (N^+^) as quantified by flowcytometry. (**C**) Difference of viral genome copy numbers in the culture supernatants collected at 0 hpi and 12 hpi as quantified by RT-qPCR. (**D**) Difference of virus infectious titers in culture supernatants collected at 0 hpi and 12 hpi as quantified by plaque assay in MARC-145 cells. Data are presented as means and SEM of the three independent experiments each of which involved four populations of PAMs collected from four different pigs. ns: non-statistically significant, * *p* < 0.05, *** *p* < 0.001, **** *p* < 0.0001.

**Figure 3 viruses-14-02822-f003:**
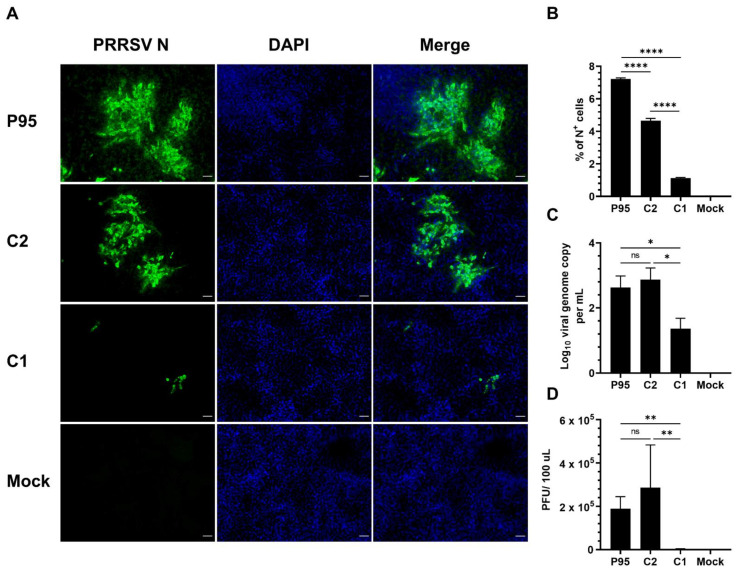
Infectivity of NCV1 P95 and its derived clones in PK15-pCD163 cells. PK15 cells stably expressing porcine CD163 (PK15-pCD163) cells were inoculated with NCV1 P95 and its derivatives C1 and C2 at an MOI of 10. Data were collected at 72 hpi. (**A**) Representative images of cells that were fixed and stained with an anti-N antibody to visualize virus-infected cells (green color). DAPI was used to stain the nuclei (blue color). Scale bar = 50 μm. (**B**) Percentage of cells expressing N protein (N^+^) as quantified by flowcytometry. (**C**) Difference of viral genome copy numbers in the culture supernatants collected at 0 hpi and 72 hpi as quantified by RT-qPCR. (**D**) Difference of virus infectious titers in culture supernatants collected at 0 hpi and 72 hpi as quantified by plaque assay in MARC-145 cells. Data are presented as means and SEM of the three independent experiments. ns: non-statistically significant, * *p* < 0.05, ** *p* < 0.01, **** *p* < 0.0001.

**Figure 4 viruses-14-02822-f004:**
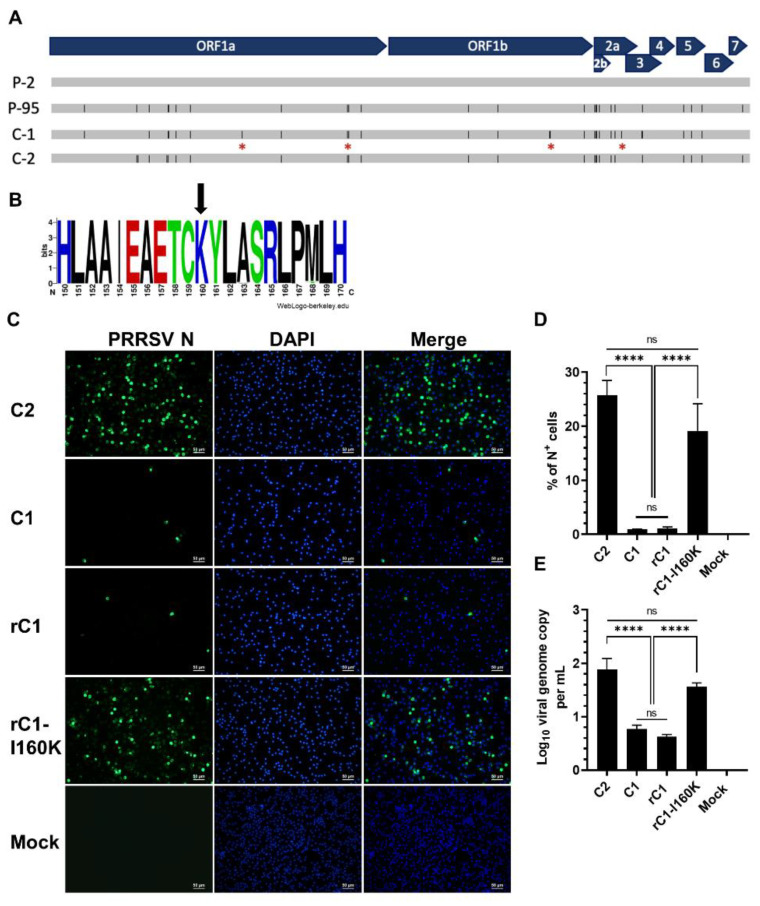
A single amino acid substitution in GP2 of C1 is responsible for impaired infectivity in PAMs. (**A**) Schematic representation of NCV1 P2, P95, C1, and C2 genomes and mutations. The virus genome organization and open reading frames are shown on top. Gray horizontal lines represent the full-length genome of indicated viruses. Black vertical bars within the gray lines depict amino acid substitutions as compared to P2. Red asterisks depict amino acid substitutions that were only found in C1. (**B**) Web logo showing the conservation of PRRSV-2 GP2 amino acid residue 150 to 170 constructed from 1467 available GenBank sequences. Web logo was constructed using the WebLogo online software (WebLogo-berkeley.edu). (**C**–**E**) Analysis of virus infectivity in PAMs at high MOI. The experiment was conducted in the same manner as described in Figure 2. (**C**) Representative images of cells that were fixed and stained with an anti-N antibody to visualize virus-infected cells (green color). DAPI was used to stain the nuclei (blue color). Scale bar = 50 μm. (**D**) Percentage of PAMs expressing N protein as quantified by flowcytometry. (**E**) Difference of viral genome copy numbers in the culture supernatants collected at 0 hpi and 12 hpi as quantified by RT-qPCR. Data are presented as means and SEM of the three independent experiments, each of which involved four populations of PAMs collected from four different pigs. ns: non-statistically significant, **** *p* < 0.0001.

**Figure 5 viruses-14-02822-f005:**
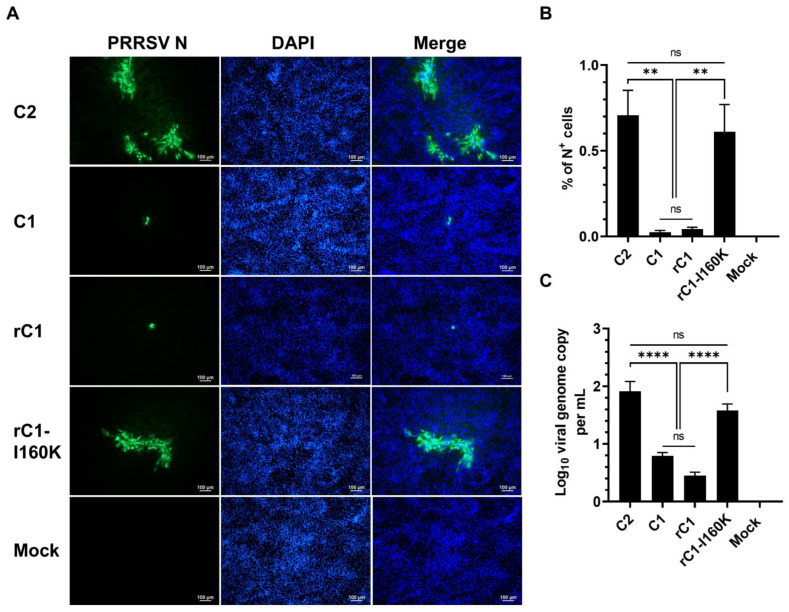
An I160K substitution in genome of C1 restores virus infectivity in PK15-pCD163 cells. Cells were inoculated with indicated viruses at an MOI of 10 and data were collected at 72 hpi in the same manner as described in Figure 3. (**A**) Representative images of cells that were fixed and stained with an anti-N antibody to visualize virus-infected cells (green color). DAPI was used to stain the nuclei (blue color). Scale bar = 100 μm. (**B**) Percentage of cells expressing N protein as quantified by flowcytometry. (**C**) Difference of viral genome copy numbers in the culture supernatants collected at 0 hpi and 72 hpi as quantified by RT-qPCR. Data are presented as means and SEM of the three independent experiments. ns: non-statistically significant, ** *p* < 0.01, **** *p* < 0.0001.

**Figure 6 viruses-14-02822-f006:**
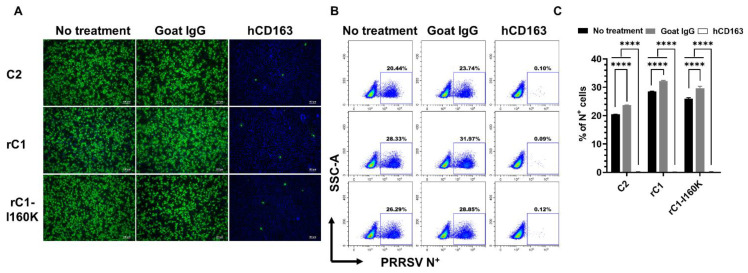
CD163 receptor is necessary for C1 virus infection. MARC-145 cells were preincubated with 10μg/100μL of either goat anti-human CD163 pAb or normal goat IgG for 1 h. Treated cells were inoculated with indicated viruses at a MOI of 2.5 TCID50 per cell and data was collected at 12 hpi. (**A**) Representative images of MARC-145 cells that were fixed and stained with an anti-N antibody to visualize virus-infected cells (green color). DAPI was used to stain the nuclei (blue color). Scale bar = 100 μm. (**B**) Representative flowcytometry plots of MARC-145 cells that were fixed and stained with a FITC-labeled anti-N antibody. Uninfected cells were used to set-up gating. (**C**) Percentage of cells expressing N protein as quantified by flowcytometry. Data are presented as means and SEM of the three independent experiments. **** *p* < 0.0001.

**Figure 7 viruses-14-02822-f007:**
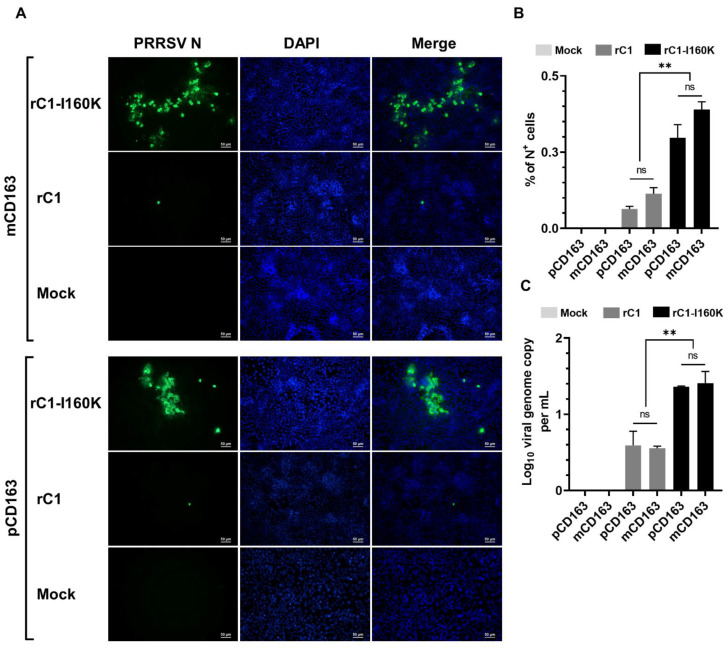
CD163 alone is not sufficient for rC1 infection. PK15 cells were transduced with lentivirus-expressing porcine (p) or monkey (m) CD163, followed by 4 passages in a selection medium to eliminate non-transduced cells. The cells were then inoculated with indicated viruses at an MOI of 10. Data were collected at 72 hpi. (**A**) Representative images of cells that were fixed and stained with an anti-N antibody to visualize virus-infected cells (green color). DAPI was used to stain the nuclei (blue color). Scale bar = 50 μm. (**B**) Percentage of cells expressing N protein as quantified by flowcytometry. (**C**) Difference of viral genome copy numbers in the culture supernatants collected at 0 hpi and 72 hpi as quantified by RT-qPCR. Data are presented as means and SEM of the three independent experiments. ns: non-statistically significant, ** *p* < 0.01.

**Figure 8 viruses-14-02822-f008:**
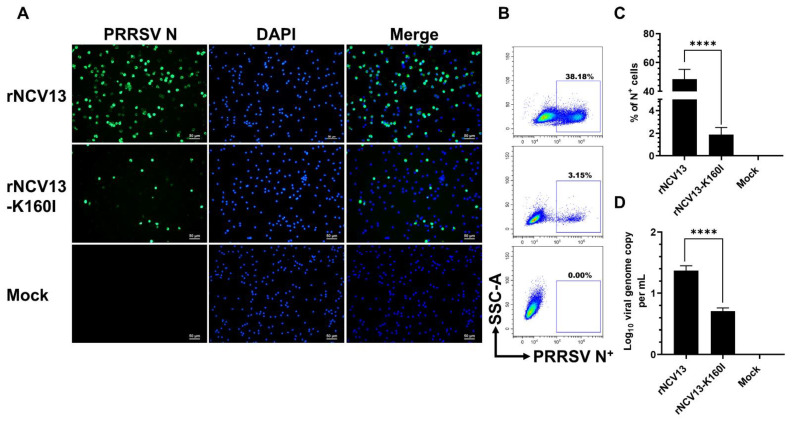
Infectivity of the low passage rNCV13 strain and its derived mutant in PAMs. PAMs collected from four piglets were inoculated with the indicated viruses at an MOI of 0.1. All data were collected at 12 hpi. (**A**) Representative images of cells that were fixed and stained with an anti-N antibody to visualize virus-infected cells (green color). DAPI was used to stain the nuclei (blue color). Scale bar = 50 μm. (**B**) Representative flowcytometry plots of PAMs that were fixed and stained with an FITC-labeled anti-N antibody. Uninfected cells were used to set-up gating. (**C**) Percentage of PAMs expressing N protein as quantified by flowcytometry. (**D**) Difference of viral genome copy numbers in the culture supernatants collected at 0 hpi and 12 hpi as quantified by RT-qPCR. Data are presented as means and SEM of the three independent experiments each of which involved four populations of PAMs collected from four different pigs. **** *p* < 0.0001.

**Figure 9 viruses-14-02822-f009:**
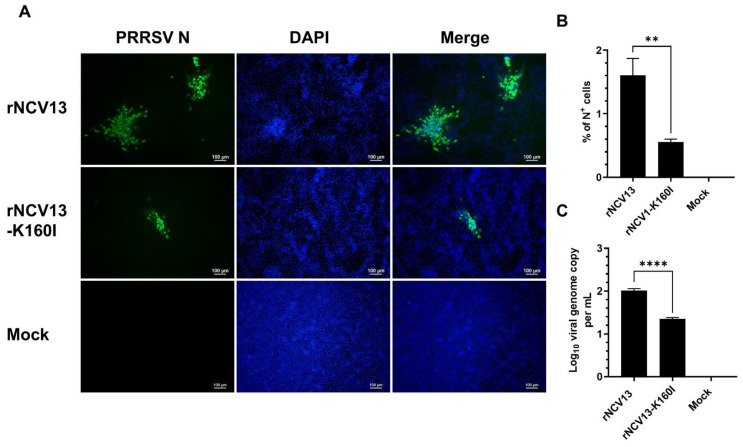
Infectivity of rNCV13 strain and its derived mutant in PK15-pCD163. PK15 cells stably expressing porcine CD163 (PK15-pCD163) cells were inoculated with the indicated viruses at an MOI of 1. Data were collected at 72 hpi. (**A**) Representative images of cells that were fixed and stained with an anti-N antibody to visualize virus-infected cells (green color). DAPI was used to stain the nuclei (blue color). Scale bar = 100 μm. (**B**) Percentage of cells expressing N protein as quantified by flowcytometry. (**C**) Difference of viral genome copy numbers in the culture supernatants collected at 0 hpi and 72 hpi as quantified by RT-qPCR. Data are presented as means and SEM of the three independent experiments. ** *p* < 0.01, **** *p* < 0.0001.

**Figure 10 viruses-14-02822-f010:**
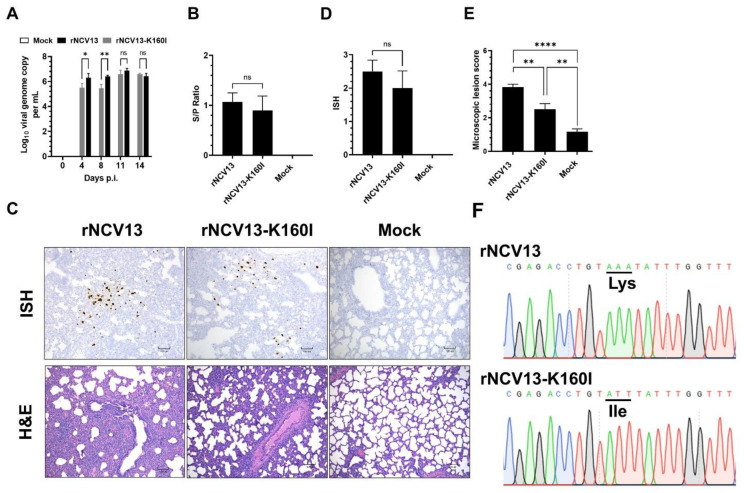
Experimental inoculation of pigs with the virulent PRRSV strain rNCV13 and its derived mutant rNCV13-K160I. (**A**) Viral load in serum samples measured by RT-qPCR. (**B**) Levels of serum antibody against PRRSV measured by ELISA. (**C**) Representative images of in situ hybridization (ISH) and H&E staining of lung sections collected at 14 dpi. (**D**) ISH scores of the lung sections. (**E**) Microscopic scores of the lung sections. (**F**) Representative sequencing chromatograms covering codon GP2 160 is underlined. Triplet AAA codes for lysine (Lys) and ATT codes for isoleucine (Ile). The viral genomes were extracted from the serum samples collected from pigs inoculated with rNCV13 or rNCV1-K60I viruses at 14 dpi and subjected to sequencing. Each nucleotide in the chromatogram is depicted with unique color code. ns: non-statistically significant, * *p* < 0.05, ** *p* < 0.01, **** *p* < 0.0001.

**Table 1 viruses-14-02822-t001:** Oligonucleotide primers used in the construction of NCV13 infectious cDNA clone.

Primer	Sequence (5′→3′)
NotI-F	GCTGCGGCCGCATGACGTATAGGTGTTGGCTC
4037R	CAAGATACAGTCTGAAACGATG
4004F	CTTAGGCTTGGCATCGTTTC
8684R	TCTTCTTCCCGCAATACTG
8096F	GTGAAGATGCTGCATTGAGAG
11997R	CATAGGATCTTCTGTAACTGCTC
11965F	GTTCACTCTGAGCAGTTACAGAAGATCCTATG
Oligo-dTR	GATGGTGAATCCGTTAGCGAGGTGTTAATTAATTTTTTTTTTTTTTTTTTTTTTTTTTTTTTTTTTTTTTTTTTTTTAATTTCGGCCGCATGGTTC

**Table 2 viruses-14-02822-t002:** Oligonucleotide primers used for the site-directed mutagenesis.

Primer	Sequence (5′→3′)	Application
11340F	CGACGTCAAAGGCACTAC	Generation of C1-I160K mutant
C1-I160K-F	AAGCCGAGACTTGTATATACTTGGCTTCCCGGCTGCC	
C1-I160K-R	TTGGCAGCCGGGAAGCCAAGTATATACAAGTCTCGGC	
P14461R	AAGGGGTTGCCGCGGAACCATCA	
14543F	TTCTGGCGTGTGCAGAGTTCTCGC	Generation of NCV13-K160I mutant
NCV13- K160I- R	AAACCAAATAAATACAGGTCTCGGCTTCAATGG	
NCV13- K160I- F	AAGCCGAGACCTGTATTTATTTGGTTTCCCGGC	
16932 R	AACGATAGAGTTTGCCCTTGGTATCC	

**Table 3 viruses-14-02822-t003:** Oligonucleotide primers used in the construction of lentivirus expressing CD163 gene.

Primer	Sequence (5′→3′)	Application
MonkeyCD163F	ACCGGCGGCCGCGCCACCATGAGCAAACTCAGAATGGTGC	Cloning of monkey CD163 in Lentivirus
MonkeyCD163R	TAGAGCTAGCTCAGTGTGCCTCAGAATGGCC	
PorcineCD163F	CCGGCGGCCGCGCCACCATGGACAAACTCAGAATGGTGC	Cloning of porcine CD163 in Lentivirus
PorcineCD163R	AGAGCTAGCTCATTGTACTTCAGAGTGGTC	

**Table 4 viruses-14-02822-t004:** Mutations only found in the C1 virus but not in the P95 and C2 viruses.

Nucleotide Position ^a^	Nucleotide Change	Protein Affected	Amino Acid Position ^b^	Amino Acid Change
4008	G→T	nsp2	1336	A→S
6240	G→A	nsp7	2080	D→N
10485	A→G	nsp11	3496	N→D
11968	A→T	GP2	160	K→I

^a^ Nucleotide positions are based on PRRSV NCV-1 genome (GeneBank accession number ON950548). ^b^ For nonstructural proteins (nsp1-nsp12), amino acid positions refer to the polyprotein pp1ab sequence of PRRSV NCV-1.

## Data Availability

Not applicable.

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
