# Peer review of "A Single Amino Acid Substitution in Porcine Reproductive and Respiratory Syndrome Virus Glycoprotein 2 Significantly Impairs Its Infectivity in Macrophages"

_viruses, 2022, doi:10.3390/v14122822_

Round 1

Reviewer 1 Report

This study provides straightforward experiments to understand the pathogenicity and infectivity of PRRSV.  The results indicated that a single mutation in GP2 could significantly affect the pathogenicity and infectivity of PRRSV, which provide an excellent approach for attenuated vaccine development. There are some questions that need to be addressed.

1. In the introduction, it is better to mention the purpose of the study.  Why do you perform the passage experiments?

2. There are three mutations in the nonstructural proteins and one K160I in GP2.  Nonstructural proteins are essential for viral replication and viral infectivity. Have you done any experiments to show the effect of mutations in nonstructural proteins on viral infectivity?

3. Will it be nice to show the viral growth curve? Does the mutation affect the viral replication at an early, late stage, or both?

4. Did the author notice that there was a size difference between the C1 and C2 viruses when you performed the plaque assay?

5. If the mutation in GP2 is critical for viral infectivity, could you introduce the mutation into the P2 virus? If the mutation significantly reduces the viral infectivity, it will shorten the whole process of attenuated vaccine development.

Author Response

We are thankful to the reviewer for their constructive suggestions on our manuscript for submission. Please see our point-by-point responses in the attached document.

Reviewer 2 Report

The manuscript describes the K160 residue in GP2 is one of the key determinants of PRRSV tropism. The manuscript is very well organised and presented, with very nice figures and tables.

However, there are some questions to explain.

1. Line319-320, “Collectively, the results clearly demonstrated that the C1 virus nearly lost its ability to infect PAMs.” However, C1 virus could infect PAMs although significantly lower than P95 and C2. Furthermore, all data were collected at 24 hpi.  

2. Similarly, line 332, “The C1 virus did not infect the PK-15 cell line stably expressing porcine CD163”. However, C1 virus could infect the PK-15 cell line stably expressing porcine CD163 in Figure3.

3. K160 residue in GP2 was found by comparing the P95 and C2. However, why not chose the infectious clone of C2 but P2 of NCV1.

4. Line427-436, “3.5. CD163 is necessary but not sufficient for the C1 virus infection” . Why chose human CD163 but not that of monkey of porcine? And “Pretreatment of MARC-145 cells with the anti-CD163 pAb completely prevented the cells from being infected with C2, rC1, and rC1-K160I viruses (Figure 6A).” Indeed, we also observed green fluorescence in Figure 6A.

5. Line368-370, “Four aa substitutions that were only found in C1 virus but not P95, or C2 included three substitutions in nonstructural proteins (A1336S, D2080N and N3496D) and one in GP2 (K160I) (Figure 4A and Table 4).” what dose the effect of three substitutions in nonstructural proteins (A1336S, D2080N and N3496D) on cell tropism? On the contrary, why analysis the effect of A462 and A718 on virus-infected PAMs or PK15-pCD163 cells.

6. In animal experiments, the pathogenic research data of different groups of pigs are few, and the monitoring of body temperature and body weight and other iconic indicators are lacking.

7. Line 434-436, Statement duplication exists Collectively, the results demonstrated that CD163 was necessary for the C1 virus to infect MARC-145 cells. Collectively, the results  demonstrated that CD163 was necessary for the C1 virus to infect MARC-145 cells“

8. There is a big difference in the positive rate between Figure 1B and Figure 1C. What causes this?

9. In Chapter 3.4 of the paper, the author compared 1467 sequences in genebank. Do the 1467 sequences cover all existing genotypes?

10. In Chapter 3.7, the author determined that the cell infection frequency of group rNCV13 was higher than that of group rNCV13-K160I by ISH assay. The number of infected cells in ISH experiment may be significantly different in different visual fields. It is suggested that RT-qPCR was used to quantify and compare the viral load in the lungs of experimental pigs.

Author Response

We are thankful to the reviewer for their constructive suggestions on our manuscript for submission. Please see point-by-point response to the reviewer's critiques in the attached document

Reviewer 3 Report

In this manuscript, the authors found and demonstrated that a K160I substitution in GP2 may greatly affect PRRSV infectivity. Well-designed experiments also provide very clear evidence. This information has great reference value and is recommended to be published on Viruses.

Author Response

We thank the review 3 for your comments. 

Round 2

Reviewer 1 Report

I have no remaining inquiries. The author addresses every question.